# OpenReview forum: "Expert-Integrated Active Learning for Optimizing LLM Agents"
_ICLR.cc/2026/Conference — ICLR 2026 Conference Withdrawn Submission_

### Official Review · Reviewer_cgP2 · 2025-10-17

**Soundness:** 2
**Presentation:** 1
**Contribution:** 1
**Rating:** 2
**Confidence:** 4

**Summary:**

```
I used LLM to fix the grammar of the Official Review, but all opinions are my own
```
Collecting expert data is challenging, which is why reinforcement learning (RL) has recently become popular. However, combining large language models (LLMs) with RL introduces several problems. One issue is that LLMs operate over a massive token vocabulary, where each token corresponds to a distinct, environment-specific action. This leads to an enormous exploration space in which only a sparse subset of tokens are actually meaningful. Another problem is that pretraining constrains the model’s exploration capability. To address this, the authors propose leveraging active learning to enhance exploration efficiency: the model first explores freely, and then the system filters for high-quality samples, which are subsequently annotated by experts for further learning. The filtering focuses on selecting challenging samples. While the idea itself is interesting, the paper’s execution appears incomplete. Many parts of the methodology are unclear, and the experiments fail to convincingly demonstrate the method’s general applicability. My current inclination is to reject, though the final decision could depend on the quality of the rebuttal.

**Strengths:**

Combining active learning with RL to guide expert labeling is interesting and potentially valuable.

**Weaknesses:**

1. I am not convinced of the generalizability of the proposed approach.
2. The paper’s exposition is too brief and lacks clarity, making it difficult to understand the key implementation details.

**Questions:**

1. Your selection rule prioritizes difficult samples and those showing little progress, but does this only work for environments that provide intermediate rewards? For benchmarks like HLE or SWE-Bench that lack intermediate signals, how would your method still apply?

2. The “expert” component seems crucial, but the paper’s description is very vague. Are these experts humans, or stronger models? Have you tested different types of experts for comparison? Could you provide concrete examples of expert demonstrations? Without examples, it’s hard to understand the actual procedure.

3. If the experts are humans, the cost would be prohibitively high. Could the process instead leverage stronger models as experts? More generally, how could you design a multi-agent RL setup with active learning using strong model-based experts?

---

> ### Author Response · Authors · 2025-11-24
> **Response to Reviewer cgP2**
>
> Thanks for your comments! We will be addressing your concerns one by one.
>
> **Weaknesses:**
>
> 1. *I am not convinced of the generalizability of the proposed approach.*
>
>    We understand your concern and have included extra validation on WebShop Benchmark. In this setup, we use almost exact hyper-parameters in AppWorld setting, with similarity threshold changed from 0.65 to 0.7, considering the similarity distribution of the benchmark dataset.
>
> | Models / Settings                                         | Cost (𝓒) | Train            |        | Test             |        |
> | --------------------------------------------------------- | -------------------- | ---------------- | ------ | ---------------- | ------ |
> |                                                           |                      | Success Rate (%) | Reward | Success Rate (%) | Reward |
> | Qwen2.5-3B-Instruct                                       | 0                    | 4.20             | 42.02  | 4.80             | 41.11  |
> | Expert Demonstration (DeepSeek-V3.2 + Oracle Information) | -                    | 35.38            | 65.71  | 34.80            | 64.34  |
> | &nbsp;&nbsp;/GRPO/baseline                                | 0                    | 52.29            | 76.42  | 56.20            | 78.04  |
> | &nbsp;&nbsp;/GRPO/full_demonstration                      | 4270                 | 66.30            | 83.31  | 68.00            | 84.74  |
> | &nbsp;&nbsp;/GRPO/active_learning                         | 970                  | 64.99            | 83.50  | 64.00            | 81.92  |
>
> The results show that with our proposed method, we can attain the around 2/3 performance growth with less than 1/4 expert demonstration cost, further validate the generalizability of our method. We have accordingly updated our paper, with result in Experiment section and more detailed experiment setup in Appendix D.
>
> 2. *The paper’s exposition is too brief and lacks clarity, making it difficult to understand the key implementation details.*
>
>    For our method, the complete pipeline is illustrated in Figure 1 and described in first subsection of Method section. For all the components in our method (Reward-based Filter, Diversity-based Selection, Mixing Strategy), we all provided designated subsections for detailed description, as well as pseudo-code for Diversity-based Selection and Mixing Strategy, respectively in main text and appendix. In our description, we use both textual description as well as mathematical expression, and all the hyper-parameters are provided in the experiment section. For the concept of expert, we provided a short description in Discussion section in our original paper and have extend the discussion with extra space permitted. We hope you can specify which part in our exposition lacks clarity as we do not get the similar feedback from other reviewers.
>
> **Questions:**
>
> 1. *Your selection rule prioritizes difficult samples and those showing little progress, but does this only work for environments that provide intermediate rewards? For benchmarks like HLE or SWE-Bench that lack intermediate signals, how would your method still apply?*
>
>    In the domain of reinforcement learning, *intermediate reward* typically refers to giving agent reward signals *before* reaching the terminal state, which in recent RL+LLM literatures is usually termed *Process Supervision*. Yet, **nowhere** in our work did we ever mention that we used Process Supervision as we follow GPRO's *default* setting with *Outcome Supervision*. We have to wonder where the reviewer get this impression from.
>    Another possible explanation is that the reviewer is refering to the "less sparse" reward design in our experiment instead of a binary reward signal, as in our work, we use the percentage of passed unit test as the reward signal. If this is the case:
>    - First, we have to make a **factual correction** in reviewer's question, as SWE-Bench, the benchmark mentioned, also uses a series of unit tests (PASS_TO_PASS & FAIL_TO_PASS) to examine the completion of tasks, which can be adopted to this reward design scheme similarly and **does not** *"lack intermediate signals"*.
>    -  More importantly, the binary reward scheme **does not** affect our reward-based filter design as we use the *average reward* over a step window to evaluate the progress. In practice, we may increase the window size for more stable evaluation for environment with binary rewards.

---

> > ### Author Response · Authors · 2025-11-24
> > **Response to Reviewer cgP2 (continued)**
> >
> > 2. *The “expert” component seems crucial, but the paper’s description is very vague. Are these experts humans, or stronger models? Have you tested different types of experts for comparison? Could you provide concrete examples of expert demonstrations? Without examples, it’s hard to understand the actual procedure.*
> >    - Expert is crucial for our framework and we apologize if there's any confusion about its definition. In our work, expert is indeed, human experts or stronger models, as we described in Discussion section.
> >    - Our work focuses on how to maximize the utilization of expert demonstration rather than the design of expert itself, therefore we only used a stronger model (DeepSeek-V3.1) as our expert. In follow-up experiments with WebShop, we also use an "enhanced" model, where we provide extra oracle information to the stronger expert model to help attain successful trajectories. We included discussion about how to perform human expert demonstration efficiently in the discussion section in original paper and have now further extended the content.
> >    - The expert demonstrations follow the identical agentic interaction process as the training agent do, since they are directly mixed into the final rollout for policy update. We have added a typical interaction process in AppWorld Benchmark in Appendix F for clarification.
> >
> > 3. *If the experts are humans, the cost would be prohibitively high. Could the process instead leverage stronger models as experts? More generally, how could you design a multi-agent RL setup with active learning using strong model-based experts?*
> >
> >    In our original paper, we **explicitly** mentioned **3 times** that, in our experiment, we use a stronger model (DeepSeek-V3.1) as the expert, respectively in the experiment setting description, the main experiment result table, and the discussion section, and it should be clear that we can leverage stronger models as experts and that the setup is executed using strong model-based experts in our experiment. Nonetheless, we have included more re-statement of this fact in multiple place with extra space permitted.
> >
> > We hope that these responses and the corresponding revisions address your concerns, and we respectfully ask you to kindly take a few minutes to revisit our paper, as the possible misunderstandings or unintentional oversights in this review does not seem to align with the current confidence score given.

---

> > > ### Comment · Reviewer_cgP2 · 2025-11-28
> > >
> > > Thank you for the detailed clarifications and the newly added experiments. After revisiting the revised submission, I agree that several of my earlier concerns came from misunderstanding your reward formulation and expert setup—I appreciate the highlighted revisions, and they helped me understand the pipeline more clearly.
> > >
> > > I do still have one question that I believe could meaningfully strengthen the paper:
> > >
> > > ### **(1) Choice of Expert Model**
> > >
> > > You currently use **DeepSeek-V3.1** as the expert. Have you explored (or do you have insights on) using a *stronger* expert model?
> > > For example:
> > >
> > > * Would a stronger expert (e.g., GPT-5 and Claude Family) significantly change the marginal benefit curve?
> > > * Is there a saturation point where expert quality stops leading to further improvement?
> > >
> > > Even a short discussion—empirical or conceptual—would help readers understand how scalable your expert-in-the-loop pipeline is.
> > >
> > > ### **(2) Self-Expert or “Best-of-Self” Demonstrations**
> > >
> > > A particularly interesting extension is treating the agent itself as an expert. This would allow your framework to turn into a **self-improving RL pipeline** (analogous to self-play), even without external models or humans.
> > > Have you considered or tested this variant?
> > > Even if not implemented, I would appreciate your thoughts on:
> > >
> > > * whether the current diversity and reward filters would still behave properly;
> > > * whether “self-expert” signals risk reinforcing existing biases;
> > > * how often pseudo-experts would need to be refreshed to avoid collapse.
> > >
> > > I believe both questions are directly aligned with the paper’s central goal, and your insight here would substantially increase the broader impact of the work.
> > >
> > > Given the clarified exposition and the additional results, I have update my score from **2 → 4**.
> > > If authors could provide deeper analysis or discussion on the expert-quality question above, I would be happy to raise my score further (**4 → 6 or 8**).

---

> > > ### Comment · Reviewer_cgP2 · 2025-11-28
> > >
> > > Hi Authors,
> > > I have tried to update my score accordingly, but it seems that the system currently does not allow me to modify the rating. I’ve already asked the AC to check whether this is a system issue or whether the scoring phase has been locked.
> > >
> > > You may also consider reaching out to the AC on your side just to confirm the status. No need to worry — once the system allows score editing again, I will update my rating based on our discussion.

---

> ### Author Response · Authors · 2025-11-28
> **Second Response to Reviewer cgP2**
>
> Thank you for your speedy response! This really means a lot for us and we are more than happy to answer these further questions you raised.
>
> ### **(1) Choice of Expert Model**
>
> *You currently use DeepSeek-V3.1 as the expert. Have you explored (or do you have insights on) using a stronger expert model? For example:*
>
> - *Would a stronger expert (e.g., GPT-5 and Claude Family) significantly change the marginal benefit curve?*
> - *Is there a saturation point where expert quality stops leading to further improvement?*
>
> *Even a short discussion—empirical or conceptual—would help readers understand how scalable your expert-in-the-loop pipeline is.*
>
> ---
>
> First, we checked whether a stronger model can provide better demonstrations for our current benchmark: We have run GPT-5.1 on AppWorld Benchmark Training set and it achieves around 66% success rate. We also calculated its Pass@1/3/5 metrics for reference. The results are shown below:
>
> | Expert        | Task Success Rate (5-run avg, %) | Pass@1 (%) | Pass@3 (%) | Pass@5 (%) |
> | ------------- | ---------------- | ---------- | ---------- | ---------- |
> | DeepSeek-V3.1 | 58.61            | 56.94      | 68.05      | 72.22      |
> | GPT-5.1       | 66.11            | 65.28      | 73.61      | 77.78      |
>
> The results show that GPT-5.1 can provides more successful trajectories overall as well as being able to demonstrate successful interactions for more tasks, so it can be said GPT-5.1 is indeed a stronger expert than DeepSeek-V3.1 on this benchmark. As our mixing strategy rejects demonstrations with lower rewards, an expert that can provide more successful trajectories in more tasks can certainly help improve the final performance of the agent.
>
> **To answer your questions:**
>
> - For the marginal benefit curve (the performance-cost tradeoff), we expect the performance differences remain small at low cost levels, but gradually widen as the cost increases, with a higher upper bound for stronger expert.
> - However, the saturation when trying to improve performance with better expert is inevitable.
> 	- First, the mixing strategy prioritize better trajectories. Since mixing strategy only allows trajectories with higher reward, at earlier stage of training, expert trajectories with diverse rewards can be sampled (training model is still weak and rollout rewards are low), while at later stages, the best-performing expert trajectories prevails (as training model gradually improves its performance and surpass those expert trajectories with lower rewards). Therefore, **even the overall success rate improves for expert, if the pass@k metrics remain similar, the final performance gain may still be limited**.
> 	- Second, **the performance of expert does not translate 100% to training agents growth**. In our experiments, there remain tasks where training agent failed to solve even when successful expert trajectories are provided. And even the model eventually converge on these successful demonstrations with more training steps, the risks of over-fitting is rather high and the improvement on train set success may not transfer well to test set.

---

> ### Author Response · Authors · 2025-11-28
> **Second Response to Reviewer cgP2 (continued)**
>
> ### **(2) Self-Expert or “Best-of-Self” Demonstrations**
>
> *A particularly interesting extension is treating the agent itself as an expert. This would allow your framework to turn into a **self-improving RL pipeline** (analogous to self-play), even without external models or humans. Have you considered or tested this variant? Even if not implemented, I would appreciate your thoughts on:*
>
> - *whether the current diversity and reward filters would still behave properly;*
> - *whether “self-expert” signals risk reinforcing existing biases;*
> - *how often pseudo-experts would need to be refreshed to avoid collapse.*
> ---
>
> This is interesting question and we did actually make some attempts during the early development stages of our work.
>
> In those experiments:
> - We used Qwen2.5-32B-Instruct as our base model, a even stronger model,
> - We used AppWorld Benchmark,
> - We implemented the expert as $\mathcal{E}(\mathcal{M}$\*$, \mathcal{I})$, where $\mathcal{M}$\* is the base model itself, and $\mathcal{I}$ is different combinations of:
> 	- a summary of the best performing trajectories for the task
> 	- the reward for the trajectories
> 	- a piece of reflection for previous trajectory
> 	- unit test descriptions and results
>
> We hoped the inclusion of these information can help model generate better trajectories for the the task, but the results were rather disappointing and no significant improvement was observed, both in expert trajectories and final agent performance. There are several reasons we suspect:
> - The ability of base model is still very limited to make meaningful reflection or improvement to its current trajectories. In the reflection generated, we observe very few fundamental and meaningful insights but more superficial comments.
> - This benchmark does not benefit from reflection or previous trajectories as much as we expected. In the expert trajectories, providing previous rollouts or reflection essentially only extends the current interaction, where expert mostly continues to repeat previous actions and get stuck on similar failures. This can be attributed to that (1) tasks from AppWorld Benchmark has rather strong sementic prior that weakens the tendency of exploration and (2) the action space are large and lack direct guidance for alternative solutions as it's implemented with codeblocks.
>
> However, this does not mean the idea of self-expert is not promising. There are works that uses self-play style strategy to improve model performance, such as MobileRL [1] mentioned in Related Work, where the task is visual mobile device manipulation, and the framework keeps top positive trajectories from previous rollouts in a replay buffer and mix them with current rollouts. This task differs from AppWorld Benchmark in that the action space is much more constrained (tap, swipe, ...), loosely connected to task goal (tap, swipe ... is more dependent on current state than initial goal) and can actually benefit from extended interaction turns (e.g., trying out all possible actions). In cases like this, self-expert is less limited by model capability and can actually explore more meaningful and successful trajectories.
>
> **To answer your questions:**
>
> - Diversity and reward filters would still apply as diversity is based on tasks rather than rollout trajectories, and reward filter is based on values calculated from rollout rewards. The main idea to select diverse tasks and attain better and more successful trajectories does not change with self-expert.
> - “Self-expert” signals indeed may increase existing biases, but it is still restrained by rollout rewards and only better-performing trajectories are selected, so the risk is limited, especially when we are already using RL methods that exploits on more successful trajectories. Nonetheless, increasing the number of self-expert samples may help reduce the biases.
> - For a cost-effective design, we can refresh self-expert replay buffer whenever no previous self-expert trajectories can pass mixing strategy (i.e., all self-expert trajectories are of same or lower rewards compared to rollouts). We have not witness collapse in our current experiments where trajectories are provided by a completely different model so we suspect the risk is very limited when self-expert samples are provided.
>
> We hope these responses help clarify your doubts. We understand our response is more conceptual rather than empirical and we will try to follow up with more experiments if time permits. Thanks again for your insightful questions as they really help us think deeper about our work!
>
> #### References
> [1] Y. Xu et al., ‘MobileRL: Online Agentic Reinforcement Learning for Mobile GUI Agents’, Oct. 24, 2025, arXiv: arXiv:2509.18119. doi: 10.48550/arXiv.2509.18119.

---

### Official Review · Reviewer_3Xpm · 2025-10-27

**Soundness:** 3
**Presentation:** 3
**Contribution:** 3
**Rating:** 4
**Confidence:** 3

**Summary:**

This paper proposes a novel framework that integrates AL and RL for training LLM–based agents in interactive environments. The authors identify the key limitation of standard RL post-training—inefficient exploration—and address it by introducing a mechanism through which the model can actively query for expert demonstrations. Specifically, the proposed system employs a reward-based filter and diversity-based task selection to identify challenging and informative tasks for expert annotation, which are then incorporated into training through a carefully designed mixing strategy. Experiments on the AppWorld benchmark show that this approach achieves notable improvements in task success rates over RL-only baselines, while requiring significantly fewer expert demonstrations compared to full supervision. The authors also provide comprehensive ablations on budget control, task diversity, and trajectory analysis, highlighting how expert data improves exploration efficiency and consistent behavioral patterns.

**Strengths:**

1. The paper presents a well-motivated approach that bridges AL with RL-based post-training. The proposed framework effectively addresses the long-standing challenge of balancing exploration efficiency and annotation cost in agentic training.
2. The framework is clearly explained, including detailed algorithmic steps for reward-based filtering, diversity-based selection, and the trajectory mixing strategy. The proposed mechanisms are conceptually sound and practically relevant.
3. The authors provide extensive experiments on AppWorld, demonstrating consistent improvements under multiple model scales (7B and 14B). The inclusion of ablation studies on similarity thresholds, early stopping, and demonstration efficiency adds strong empirical support.

**Weaknesses:**

1. The proposed method still requires non-trivial manual effort. Although active selection reduces the annotation cost, the framework's practicality in large-scale real-world deployment remains uncertain. It would be beneficial to discuss possible automation or self-improvement mechanisms to reduce expert reliance.
2. The experiments are conducted solely on AppWorld. While this benchmark is well-suited for interactive environments, validation on additional settings (e.g., WebShop, OSWorld, or τ-bench) would strengthen the claim of general applicability.
3. The paper uses synthetic expert data (generated by DeepSeek-V3.1) instead of real human annotations. This design choice raises questions about the robustness of the findings in true human-in-the-loop scenarios.
4. The framework involves repeated task selection, similarity computation, and trajectory mixing. The paper does not provide detailed analysis of computational overhead or memory usage, which would be important for evaluating its scalability to larger models or environments.

**Questions:**

Please check my comments in Weaknesses.

---

> ### Author Response · Authors · 2025-11-24
> **Response to Reviewer 3Xpm**
>
> Thanks for your comments! We will be addressing your concerns one by one.
>
> **Weaknesses:**
>
> 1. *The proposed method still requires non-trivial manual effort. Although active selection reduces the annotation cost, the framework's practicality in large-scale real-world deployment remains uncertain. It would be beneficial to discuss possible automation or self-improvement mechanisms to reduce expert reliance.*
>
>    In our original paper, we included a short discussion about the implementation of expert that can reduce the distribution drift, increase the data utilization. With the addition of one new page allowed in the rebuttal phase, we have provided a formal and more detailed description about the possible implementation of the expert. In practice, the expert design can be more complex, efficient and capability-aware with joint effort of human expert, LLM and task-specific information, which may include current rollouts from agent, oracle information and evaluation metrics. This gives many possible implementations that can improve performance while reducing cost:
> - Expert trajectories can be formed by human expert supervising the model's interaction with the environment and making necessary adjustments at each turn. The model can be either the external model or the current training model. This interaction helps reduce human labor while maintaining consistency between expert trajectories and model rollouts.
> - Expert trajectories can be formed via model reflection. Providing current rollouts in context increases the likelihood of sampling successful trajectories with LLM or building accumulative progress based on the capability of the current training agent. Oracle information and evaluation metrics can further reduce the difficulty of obtaining successful trajectories.
> - Providing current rollouts and evaluation metrics helps the human expert design trajectories with specific rewards or environment feedback, informed by the mixing strategy, improving the efficiency of expert trajectory generation.
>
>
> 2. *The experiments are conducted solely on AppWorld. While this benchmark is well-suited for interactive environments, validation on additional settings (e.g., WebShop, OSWorld, or τ-bench) would strengthen the claim of general applicability.*
>
>    This is a great suggestion and we have followed and test our method on WebShop. In this setup, we use almost exact hyper-parameters in AppWorld setting, with similarity threshold changed from 0.65 to 0.7, considering the similarity distribution of the benchmark dataset.
>
> | Models / Settings                                         | Cost (𝓒) | Train            |        | Test             |        |
> | --------------------------------------------------------- | -------------------- | ---------------- | ------ | ---------------- | ------ |
> |                                                           |                      | Success Rate (%) | Reward | Success Rate (%) | Reward |
> | Qwen2.5-3B-Instruct                                       | 0                    | 4.20             | 42.02  | 4.80             | 41.11  |
> | Expert Demonstration (DeepSeek-V3.2 + Oracle Information) | -                    | 35.38            | 65.71  | 34.80            | 64.34  |
> | &nbsp;&nbsp;/GRPO/baseline                                | 0                    | 52.29            | 76.42  | 56.20            | 78.04  |
> | &nbsp;&nbsp;/GRPO/full_demonstration                      | 4270                 | 66.30            | 83.31  | 68.00            | 84.74  |
> | &nbsp;&nbsp;/GRPO/active_learning                         | 970                  | 64.99            | 83.50  | 64.00            | 81.92  |
>
> The results show that with our proposed method, we can attain the around 2/3 performance growth with less than 1/4 expert demonstration cost, further validate the generalizability of our method. We have accordingly updated our paper, with result in Experiment section and more detailed experiment setup in Appendix D.
>
> 3. *The paper uses synthetic expert data (generated by DeepSeek-V3.1) instead of real human annotations. This design choice raises questions about the robustness of the findings in true human-in-the-loop scenarios.*
>
> The synthetic expert data should not invalidate the findings in true human-in-the-loop scenarios. In our framework, the whole process operates independently of the expert design, and expert can only affect the final performance with the demostration it provides. In our experiment, we prove that a stronger model can provides the demonstrations needed to improve the performance, so human experts, which should be much more capable than any models in corresponding domain, should also help achieve, if not further enhancing, the improvement inthe agent's performance. We also include more discussion about possible expert design in the updated Discussion section, which involves the complex of human expert $\mathcal{H}$, LLM $\mathcal{M}^*$ and task-specific information $\mathcal{I}_i$ .

---

> > ### Author Response · Authors · 2025-11-24
> > **Response to Reviewer 3Xpm (continued)**
> >
> > 4. *The framework involves repeated task selection, similarity computation, and trajectory mixing. The paper does not provide detailed analysis of computational overhead or memory usage, which would be important for evaluating its scalability to larger models or environments.*
> >
> > This is a great suggestion. In one sentence, the overhead computational and memory usage is almost neglectable compared to the training process. The computational cost mostly comes in:
> >
> >    - Reward-based Filter: In this module, we calculate the average reward of tasks in two concessive step windows. We first iterate over each rollout to get the rewards, then calculate the difference in average reward for each task involved. The time complexity is $O(u\times G \times |\mathcal{D}_n|)$, where $u/2$ is the size of a window, $G$ is the rollout number for each task and $|\mathcal{D}_n|$ is the number of tasks at step $n$. The memory complexity is $O(|\mathcal{D}_n|)$. The operations are simple scalar additions and multiplications and should cost minimal time or memory.
> >    - Diversity-based Selection:
> >      - In this module, we use simlarity between tasks to further select appropriate expeert tasks. The similarity computation cost is very dependent on the design of the similarity metric design. In our experiment, we use the similarity of the set of API functions involved in tasks, which can be pre-computed with time complexity and memory complexity $O(|\mathcal{D}|^2)$. The operations are simple set similarity calculation ($|A \cap B| / | A \cup B |$) and should cost minimal time or memory.
> >      - The time and memory complexity of Max-Min selection is $O(|\hat{\mathcal{D}}_n|^3)$ and $O(|\hat{\mathcal{D}}_n|^2)$ in theory, where $|\hat{\mathcal{D}}_n|$ is the number of candidate expert tasks at step $n$, but much less in practice as we apply a threshold to end it prematurely. For a larger batch size, we can further reduce the cost by using a sampling-based Max-Min selection where instead of calulating the similarity of all the tasks left, we sample $k$ tasks to check if there's any tasks that qualifies, which further reduce the time complexity to $O(|\hat{\mathcal{D}}_n|^2\times k)$.
> >    - Trajectory Mixing: In this module, we mix the agent rollouts with expert demonstrations with a few pre-defined rules. The time and memory complexity is $O(G+m)$, where $G$ is the rollout number for each task and $m$ is the number of expert demonstrations for each tasks, which are all rather small constants.
> >    In conclusion, the extra operations introduced in these modules are **orthogonal to the size of the models or environments** and **mostly only related to the rollout batch size**. More importantly, these operations are almost all scalar operations is neglectable compared to LLM operation during the rollout process itself. We have included this discussion in our appendix.
> >
> > We hope that these responses and the corresponding revisions address your concern and are willing to answer any further questions you may have!

---

### Official Review · Reviewer_BP7A · 2025-11-01

**Soundness:** 4
**Presentation:** 4
**Contribution:** 4
**Rating:** 2
**Confidence:** 3

**Summary:**

This paper proposes an expert-integrated active learning (AL) framework to optimize LLM agents through GRPO-based RL post-training. The core approach involves a two-stage task selection process: (1) reward-based filtering to identify tasks with persistent failure and stagnant returns; (2) diversity-based max-min selection to eliminate redundancy. Subsequently, expert demonstrations are requested only for these tasks. Training employs a hybrid policy that blends high-quality expert trajectories with model self-sampled trajectories at a constrained ratio, leveraging expert knowledge while preserving exploration capabilities.

**Strengths:**

## Strengths

1. **Originality**
- The paper integrates active selection of tasks with expert demonstrations into an on-policy GRPO training loop for agents—moving beyond passive offline demo mixing; the reward-stagnation + success-rate filter is a simple, RL-appropriate proxy for uncertainty.
- The diversity-based max-min selection with a historical buffer is a practical twist that avoids repeatedly annotating near-duplicates across steps.
2. **Quality**
- Under matched budgets and consistent evaluation protocols, the method demonstrates stable and statistically significant performance gains relative to baselines, reflecting its empirical quality.
- Not only did the report show overall improvement, but it also presented detailed metrics analysis such as demonstration utilization rate and reuse frequency (efficiency), revealing the phenomenon that "a small number of high-value demonstrations are repeatedly utilized." This aligns with the expectation that active learning reduces annotation overhead.

3. **Clarity**
-  The paper is well-organized, with a coherent narrative from motivation to validation, making its technical and empirical contributions easy to follow.

4. **Significance**
- The paper addresses a real bottleneck for agent RL—inefficient exploration and annotation cost—and shows measurable gains on a recognized agentic benchmark with reduced cost


---

**Weaknesses:**

## Weaknesses
1. Lack of side-by-side comparison with existing AppWorld proxy methods: The experiment only compared the author's own three settings (GRPO/baseline, full demo, active learning) without providing direct numerical comparisons against representative methods publicly reported on AppWorld.
2. Figures and tables are not self-contained: beyond titles, they lack descriptive captions specifying the evaluation setup, metric definitions/units,  and significance markers. This weakens clarity and makes it hard to interpret results at a glance.



---

**Questions:**

## Questions
1. Would you add head-to-head comparisons against representative AppWorld agents reported in prior work (e.g., recent public baselines), under the same evaluation protocol?
2. In the main results table, why are Test-set results for the expert model (DeepSeek-V3.1)—which supplies demonstrations—omitted?

---

> ### Author Response · Authors · 2025-11-24
> **Response to Reviewer BP7A**
>
> Thanks for your comments! We will be addressing your concerns one by one.
>
> **Weaknesses:**
> 1. *Lack of side-by-side comparison with existing AppWorld proxy methods: The experiment only compared the author's own three settings (GRPO/baseline, full demo, active learning) without providing direct numerical comparisons against representative methods publicly reported on AppWorld.*
>
> As mentioned in our original paper, our GRPO implementation is based on LOOP as the baseline, which is the current SOTA RL method on AppWorld benchmark (on Qwen2.5-32B-Instruct), with a few sampling techniques to stablize the process. Nonetheless, it might be helpful to introduce some other baseline method to help understand the experiment results. Here we provide two more baselines for comparison:
>    - */SFT/full_demonstration*: A setting where we directly finetune model on expert demonstrations of all tasks,
>    - */GRPO⁰/baseline*: A setting where we use vanilla GRPO, with normalized reward and without expert demonstrations.
>
> | Models / Settings          | Cost (𝓒) | **Train** Task SR (%) | Train Scene SR (%) | **Test Normal** Task SR (%) | Test Normal Scene SR (%) | **Test Challenge** Task SR (%) | Test Challenge Scene SR (%) |
> | -------------------------- | --------- | --------------------- | ------------------ | --------------------------- | ------------------------ | ------------------------------ | --------------------------- |
> | Qwen2.5-7B-Instruct        | 0         | 1.38                  | 0.00               | 0.60                        | 0.00                     | 1.92                           | 0.00                        |
> | Qwen2.5-14B-Instruct       | 0         | 23.61                 | 8.33               | 10.71                       | 1.79                     | 6.00                           | 1.44                        |
> | Expert (DeepSeek-V3.1)     |           | 56.94                 | 33.33              | 56.55                       | 37.50                    | 40.63                          | 18.75                       |
> | **Qwen2.5-7B-Instruct**    |           |                       |                    |                             |                          |                                |                             |
> | /GRPO⁰ / baseline          | 0         | 40.27                 | 29.17              | 10.71                       | 5.36                     | 5.27                           | 1.44                        |
> | /SFT / full_demonstration  | 360       | 44.44                 | 25.00              | 27.98                       | 10.71                    | 8.39                           | 2.16                        |
> | /GRPO / baseline           | 0         | 41.67                 | 33.33              | 11.90                       | 3.57                     | 3.60                           | 0.72                        |
> | /GRPO / full_demonstration | 360       | 72.22                 | 50.00              | 29.76                       | 16.07                    | 10.07                          | 2.16                        |
> | /GRPO / active_learning    | 165       | 66.67                 | 45.83              | 28.57                       | 12.50                    | 7.19                           | 1.44                        |
> | **Qwen2.5-14B-Instruct**   |           |                       |                    |                             |                          |                                |                             |
> | /GRPO⁰ / baseline          | 0         | 69.44                 | 50.00              | 43.45                       | 26.79                    | 19.64                          | 8.63                        |
> | /SFT / full_demonstration  | 360       | 56.94                 | 37.50              | 37.50                       | 17.86                    | 14.39                          | 3.60                        |
> | /GRPO / baseline           | 0         | 72.22                 | 58.33              | 43.45                       | 32.14                    | 18.94                          | 7.91                        |
> | /GRPO / full_demonstration | 360       | 76.39                 | 66.67              | 51.19                       | 30.36                    | 23.50                          | 7.91                        |
> | /GRPO / active_learning    | 75        | 77.78                 | 66.67              | 49.40                       | 32.14                    | 21.82                          | 9.35                        |
>
> The results show that */SFT/full_demonstration* are less competent than */GRPO/full_demonstration*, which use same number of expert demonstrations, while */GRPO⁰/baseline* and */GRPO/baseline* achieves similar result, indicating LOOP's advantage maybe more significant with larger models.

---

> ### Author Response · Authors · 2025-11-24
> **Response to Reviewer BP7A (continued)**
>
> 2. *Figures and tables are not self-contained: beyond titles, they lack descriptive captions specifying the evaluation setup, metric definitions/units, and significance markers. This weakens clarity and makes it hard to interpret results at a glance.*
>
> In our original paper, we have provided the evaluation setup at the beginning the our Experiment section. The metrics are defined at Method section as well as Experiment section where the figures are cited. The necessary units are already included in tables and figures and those unattended are unitless values, mostly indicating the number of occurrences or instances. With the addition of one new page allowed in rebuttal phase, we have restated these facts in corresponding captions for better clarity.
>
> **Questions:**
> 1. *Would you add head-to-head comparisons against representative AppWorld agents reported in prior work (e.g., recent public baselines), under the same evaluation protocol?*
>
>  See our response to Weaknesses 1.
>
> 2. *In the main results table, why are Test-set results for the expert model (DeepSeek-V3.1)—which supplies demonstrations—omitted?*
>
> The test-set results initially seemed unnecessary to us, as the main focus of the section is the comparison among the _/GRPO/baseline_, _/GRPO/full_demonstration_, and _/GRPO/active_learning_ settings, independent of the expert model’s performance. However, including the test results can indeed improve the clarity of the presentation, and we have added them to the table accordingly (see the table above).
>
> We hope that these responses and the corresponding revisions will help in re-evaluating our work, as we find it somewhat difficult to understand how the final score aligns with the subscores and the content of the review. We wonder if there is any misunderstanding and are willing to answer any further quesiton you may have!

---

### Author Response · Authors · 2025-11-24
**Manuscript Update**

With the constructive suggestion from reviewers, we have made the following revisions to our original paper, highlighted with bright yellow:
## Experiment Issues
- We have provided two new baseline method for comparison in AppWorld benchmark, */SFT/full_demonstration* and */GRPO⁰/baseline*.
- We have included extra experiments on WebShop benchmark to further validate our proposed method.
- We have included the expert's performance on benchmark test sets.
## Content Issues
- We have extend the discussion about expert design with human expert $\mathcal{H}$, LLM $\mathcal{M}^*$ and task-specific information $\mathcal{I}_i$ in Discussion section.
- We have included the discussion about computation and memory cost in Appendix E.
- We have included an example trajectory for reference in Appendix F.
## Presentation Issues
- Figures and tables are now provided with more detailed description.

---

### Author Response · Authors · 2025-11-27
**Request for Feedback to Proceed with Rebuttal**

Dear Reviewers,

We hope you are doing well. As the rebuttal period is currently ongoing, we are writing to kindly inquire whether you could share your reviews on our response at your earliest convenience. This would greatly help us prepare timely and meaningful further responses to any more questions or concerns you may have.

We sincerely appreciate your time and efforts.

Best regards,
The Authors

---

### Comment · Reviewer_cgP2 · 2025-11-28

Hi AC,
I would like to adjust my score for this submission, but it seems that the system no longer allows me to modify the rating. Could you please check if this is a system restriction or if scoring has already been locked?
Thank you!

---

### Author Response · Authors · 2025-12-01
**Withdrawal Statement**

Dear Area Chair and Reviewers,

After careful consideration, we have decided to withdraw this submission. During the rebuttal phase, we believe we have answered all questions from the reviewers, but received little feedback in return, especially after the widespread anonymization bug that leads to the disabling of commenting functionality. We wish to clarify that at no stage during the rebuttal did we breach the double-blind anonymity policy by checking reviewer/AC's identity or making contact outside of the reviewing system, and the withdrawal is solely due to our concerns about the inability to properly communicate with reviewers and obtain a fair, fully informed assessment. We sincerely thank the Area Chair and reviewers for their time and effort.

Best regards,

Authors

---

### Note · Authors · 2025-12-01

I have read and agree with the venue's withdrawal policy on behalf of myself and my co-authors.